# Retinitis Pigmentosa: Novel Therapeutic Targets and Drug Development

**DOI:** 10.3390/pharmaceutics15020685

**Published:** 2023-02-17

**Authors:** Kevin Y. Wu, Merve Kulbay, Dana Toameh, An Qi Xu, Ananda Kalevar, Simon D. Tran

**Affiliations:** 1Division of Ophthalmology, Department of Surgery, University of Sherbrooke, Sherbrooke, QC J1G 2E8, Canada; 2Faculty of Medicine, University of Montreal, Montreal, QC H3T 1J4, Canada; 3Faculty of Medicine, McGill University, Montreal, QC H3G 2M1, Canada; 4Faculty of Dental Medicine and Oral Health Sciences, McGill University, Montreal, QC H3A 1G1, Canada

**Keywords:** retinitis pigmentosa, ER stress, retinal degeneration, photoreceptor cell death, therapeutic target, neuroprotection, optogenetics, gene therapy, stem cell therapy, preclinical studies

## Abstract

Retinitis pigmentosa (RP) is a heterogeneous group of hereditary diseases characterized by progressive degeneration of retinal photoreceptors leading to progressive visual decline. It is the most common type of inherited retinal dystrophy and has a high burden on both patients and society. This condition causes gradual loss of vision, with its typical manifestations including nyctalopia, concentric visual field loss, and ultimately bilateral central vision loss. It is one of the leading causes of visual disability and blindness in people under 60 years old and affects over 1.5 million people worldwide. There is currently no curative treatment for people with RP, and only a small group of patients with confirmed RPE65 mutations are eligible to receive the only gene therapy on the market: voretigene neparvovec. The current therapeutic armamentarium is limited to retinoids, vitamin A supplements, protection from sunlight, visual aids, and medical and surgical interventions to treat ophthalmic comorbidities, which only aim to slow down the progression of the disease. Considering such a limited therapeutic landscape, there is an urgent need for developing new and individualized therapeutic modalities targeting retinal degeneration. Although the heterogeneity of gene mutations involved in RP makes its target treatment development difficult, recent fundamental studies showed promising progress in elucidation of the photoreceptor degeneration mechanism. The discovery of novel molecule therapeutics that can selectively target specific receptors or specific pathways will serve as a solid foundation for advanced drug development. This article is a review of recent progress in novel treatment of RP focusing on preclinical stage fundamental research on molecular targets, which will serve as a starting point for advanced drug development. We will review the alterations in the molecular pathways involved in the development of RP, mainly those regarding endoplasmic reticulum (ER) stress and apoptotic pathways, maintenance of the redox balance, and genomic stability. We will then discuss the therapeutic approaches under development, such as gene and cell therapy, as well as the recent literature identifying novel potential drug targets for RP.

## 1. Introduction

Retinitis pigmentosa (RP) is a heterogeneous group of hereditary diseases that lead to the degeneration of the retina’s photoreceptor cells, starting with the rods, resulting in a gradual loss of vision over time. It is the most common type of inherited retinal dystrophy and has a high burden on both patients and society. It is one of the leading causes of visual disability and blindness in people under 60 years old and affects over 1.5 million people worldwide [1]. Despite the heterogeneous phenotypes of the diseases, common symptoms of RP are nyctalopia and gradual loss of peripheral vision leading to blindness.

## 2. Overview of Retinitis Pigmentosa

### 2.1. Clinical Manifestations

Effective management of Retinitis Pigmentosa (RP) begins with a thorough diagnostic evaluation. RP diagnosis is based on clinical manifestations such as nyctalopia, peripheral visual field loss, and characteristic fundus changes, and is confirmed by abnormal ERG results. Typical fundus changes include bone spicules hyperpigmentation and hypopigmentation, waxy disc pallor, and arteriolar narrowing. There are also two well-recognized atypical fundus phenotypes: retinitis punctata albescens and choroideremia [2].

### 2.2. Classification

RP can be classified into two categories: syndromic and non-syndromic.

Non-syndromic RP, involving only retina dystrophy without any other organ being affected, has a prevalence of 1:5000 [3]. Retinitis Pigmentosa (RP) can be caused by sporadic mutations, which are the most frequent etiology, but genetic predisposition is still the primary risk factor. The mode of inheritance can be classified as autosomal dominant (occurring in 30–40% of cases), autosomal recessive (occurring in 50–60% of cases), or X-linked (occurring in 5–15% of cases) [4]. Therefore, family history and genetic testing are important tools in clarifying the inheritance pattern and determining the risk for RP.

Syndromic RP includes Leber Congenital Amaurosis (LCA), Usher syndrome, and Bardet-Biedl syndrome. LCA is mostly an autosomal recessive hereditary condition caused by RPE65 mutation and characterized by early vision loss (from 20/200 to no light perception), congenital abnormal pupillary response, and nystagmus noticed in infancy. Usher syndrome, the most frequent syndromic form of RP (prevalence of 3:100,000), involves classic RP clinical presentation and different levels of auditory and vestibular dysfunction depending on the subtype [5]. Bardet-Biedl syndrome is the second most frequent syndromic form of RP (prevalence of 1:160,000 in Northern Europeans) [6] (pp. 117–136). It is an autosomal recessive hereditary disease caused by BBS1-BBS21 gene mutations (predominantly BBS1) with multisystemic presentation including polydactyly, genital abnormality, cognitive impairment, and classic RP symptoms occurring within the first decade of life.

While there are several rare forms of syndromic RP, only two types are clinically significant due to the availability of treatments that can preserve vision. Bassen–Kornzweig syndrome is an autosomal recessive disorder that causes retinal and neurological degeneration and is characterized by deficiencies of vitamin A and vitamin E [7]. Early intervention of vitamin A (300 IU/kg/day) and vitamin E (100 IU/kg/day) supplements was reported effective in slowing retinal degeneration [8]. In Refsum disease, visual decline can be controlled through weight control and restriction of foods containing phytanic acid, as accumulation of this acid due to a defective enzyme is the main cause of retinal and neurological degeneration [9].

### 2.3. Management and Prognosis

The prognosis of RP is difficult to establish because of the heterogeneity of gene mutations. The progression of the disease can vary depending on the specific gene mutations and other factors. Some studies showed that the onset of symptoms could be in childhood or in adulthood, with an average annual progression of 4–12% visual field loss [10]. In general, autosomal dominant RP has the least severe vision loss, whereas X-linked RP has the most severe manifestations and the worst prognosis [1]. Progression of visual field loss in RP typically begins with sectorial scotoma in the mid-peripheral areas, advancing to partial ring scotoma, complete ring scotoma, and ultimately resulting in total blindness. Standard medical follow-up for patients with RP includes annual ophthalmic examination, which includes measurement of visual acuity and Goldmann visual field, dilated fundoscopy, optical coherence tomography (OCT), and occasionally fluorescein angiography (FA). While a and b wave amplitude decline in electroretinogram (ERG) is a sensitive tool to assess the progression of RP, it is not always necessary for annual follow-up [2].

Although most of the RP patients will become legally blind in their fourth decade of life, they will not be completely blind as they still have some macular function remaining [2]. At the stage of panretinal dystrophy, common signs observed include optic nerve head drusen, cystoid macular edema, vitreous cells, epiretinal membranes, and posterior subcapsular cataracts. Diminished visual acuity is a common symptom experienced by patients with RP and is typically caused by the complications of cystoid macular edema and posterior capsular opacification.

There is currently no curative treatment for most patients with RP. Only a small portion of RP patients with RPE65 gene mutation are eligible to receive the target gene therapy. Most patients with RP rely on conventional treatment options, including vitamin A supplements, protection from sunlight, visual aids, and medical and surgical interventions. These interventions aim to manage symptoms, prevent ophthalmic complications, and slow the progression of the disease, but do not cure the disease.

## 3. Conventional Treatments and Limitations

### 3.1. Dietary Supplements (Vitamin A, DHA, Lutein) [11]

Vitamin A is a fat-soluble vitamin stored mostly in the liver in the form of retinyl ester. It is well known that different forms of vitamin A (all-trans-retinol) play an essential role in visual cycle, in retinal pigment epithelium cell metabolism, and in phototransduction [12]. Berson conducted multiple randomized clinical trials on RP patients to study the effectiveness of vitamin A, DHA, and lutein as treatments. The results showed that vitamin A alone may slow the decline of ERG amplitude but did not show a significant difference in visual field area or visual acuity [13]. DHA did not show a significant difference when combined with vitamin A. However, a subgroup analysis of DHA showed a slower visual field decline [14]. Lutein combined with vitamin A only showed a slower rate of decline in the total point score for the HFA 60-4 program, as a secondary outcome [15]. Despite mixed results, the combination of vitamin A, lutein, and fish oil containing DHA is still recommended as therapy for RP patients. However, Rayapudi et al., 2013 performed a systematic review of three clinical trials (Berson 1993, Berson 2004, Hoffman 2004), concluding that there was no evidence to support vitamin A, DHA, or a combination thereof providing RP patients with significant benefit [12]. Furthermore, regular use of high doses of Vitamin A can cause a range of side effects and complications. These may include short-term side effects (e.g., nausea, loss of appetite, headaches, dizziness, fatigue, dry and itchy skin), as well as long-term complications (e.g., liver toxicity, increased risk of osteoporosis and hip fractures) [16]. The risk of teratogenicity associated with vitamin A supplements is particularly concerning for the typical RP population, as many patients are in their young reproductive years. For all the reasons above, vitamin A treatment remains controversial. However, there is evidence supporting its effectiveness for a small, genetically distinct subgroup of RP patients with PRPH2-associated retinitis pigmentosa [17].

### 3.2. Cystoid Macular Edema (CME) Treatment

Cystoid macular edema (CME) can be a complication in up to 38% of RP cases, which can lead to decreased visual acuity [18,19]. According to the systematic review of Bakthavatchalam 2017, it is well shown in multiple studies that the benefits of oral carbonic anhydrase inhibitors (acetazolamide and methazolamide) and topical carbonic anhydrase inhibitors (dorzolamide and brinzolamide) are significant as first-line treatments. For patients not responding to carbonic anhydrase inhibitors, second-line treatments, including intravitreal steroids injection, oral corticosteroid, anti-VEGF injection, as well as topical or local non-steroidal anti-inflammatory drug (NSAID), are also proven as effective pharmaceutical therapies [20].

### 3.3. Protection from Sunlight

The retina is vulnerable to oxidative stress due to its high metabolic rate and oxygen consumption, as well as the presence of photosensitizer molecules in the photoreceptors that are constantly exposed to light and oxidative stress. This leads to the accumulation of reactive oxygen species (ROS) in the retinal pigment epithelium (RPE). Multiple pathways have shown that oxidative microglial activation may perpetuate a cycle of neuroinflammation and degeneration in RP [21]. A study on animal models of RP suggested that absence of light exposure was associated with reduction in the rate of photoreceptor degeneration [22]. Increased housing light intensity for rd10 mice was shown to accelerate retinal degeneration by activating cell death, oxidative stress pathways, and inflammatory cells. Therefore, light protection may be a potential intervention to slow down the progression of the disease in some cases of RP [23]. However, one case report noted that an RP patient with mono-ocular occlusion for over 40 years had an equivalent fundus in both eyes. Thus, there is a lack of convincing studies to confirm the hypothesis on whether sunlight deprivation slows down retinal degeneration in RP.

### 3.4. Visual Aids

Some visual aids could help to improve patients’ quality of life. For example, night vision pocketscopes, goggles, or other devices that could amplify light would help to alleviate night blindness [24,25,26]. A study conducted by Ikeda et al. (2015) demonstrated the effectiveness of a research device specifically designed to assist patients with RP-induced night blindness. The device was equipped with a camera that provided a minimum illuminance of 0.08 lux, and the subjects using the device achieved significantly higher success rates in completing a walking task in dimly lit rooms [27].

### 3.5. Surgical Intervention

For end-stage RP patients with bare light perception, ARGUS II prothesis is an option. The implant is an epiretinal electrode chip that stimulates the retina with electrical impulses. There is a video recorder mounted on glasses capturing video images, converting them to electrical impulses, and transmitting them to the implant. This device allows patients with profound vision loss to see lines and edges of surrounding objects [28]. Implantation of an epiretinal device consists of invasive surgery, and only limited RP patients are candidates. In a phase II clinical trial of the ARGUS II, although participants had significantly better scores in all visual function tests when the device was in use, more than one-third of the participants experienced serious adverse events related to the device or surgery, such as conjunctival erosion, conjunctival dehiscence, hypotony, and endophthalmitis. In addition, ARGUS II prothesis at its current stage is only beneficial for end-stage RP patients in allowing them to restore minimal vision. RP patients with mild visual impairment would not benefit from this device to restore normal vision [29]. However, a recent study has found that the safety profile of the Argus II has significantly improved compared to the pre-approval phase, with no significant issues reported up to four years post-implantation [30].

In short, the conventional treatments listed above all have limitations. Additionally, most of them do not target the underlying pathogenesis of RP.

## 4. Recent Therapeutic Advances

Gene therapy is one of the promising avenues for the treatment of RP as it targets the underlying genetic causes of the condition.

At present, Luxturna (voretigene neparvovec) is the only approved gene therapy for RP, and it is only authorized for the treatment of a small sub-population of patients who have the RPE65 gene mutation, which represents 0.3–1% of all RP cases. The RPE65 gene is responsible for vitamin A metabolism in the visual cycle, and its mutation causes the syndrome of Leber congenital amaurosis (LCA). In a phase III clinical trial, 31 patients with confirmed biallelic RPE65 mutations were treated with voretigene neparvovec, an adeno-associated virus (AAV2) vector containing modified human RPE65, resulting in significant visual function improvement and no serious adverse events after one year and durability of improvement after three to four years follow-up [31,32,33]. The success of this gene therapy has inspired further research targeting other gene mutations associated with RP. Numerous clinical trials of potential gene therapy for retinitis pigmentosa are currently ongoing and can be found on clinicaltrials.gov.

The following sections will focus on novel therapeutic targets in the preclinical phase. We will discuss the molecular pathways involved in the development of retinitis pigmentosa (RP) and the various preclinical therapeutic approaches currently under development, including gene therapy, cell therapy, optogenetics, neuroprotective agents, exosome therapy, and novel treatment targets identified in recent literature (Figure 1).

## 5. Gene Therapy

### 5.1. Overview of Gene Therapy Methods

The field of gene therapy has tremendously evolved over the past few years, especially in the treatment of inherited retinal diseases [34]. There are mainly two approaches for gene therapy that can be applied to RP, influenced by the inheritance pattern of the disease. In recessive RP, where a loss of function of the interest protein is displayed, the aim is to take a gene complementation approach. In dominant RP, gene therapy approaches involve gene suppression with or without gene complementation.

To achieve the therapeutic goal of RP management, numerous gene therapy methods have been developed using viral or non-viral vectors. In this section, we will review the main gene therapy methods currently under study for the development of therapeutic approaches for RP management.

#### 5.1.1. Viral Vectors

Viral vectors for gene therapy in the treatment of RP have been widely studied, several of them reaching clinical studies [35]. There are mainly three types of vector strategies based on viruses: adenoviruses (Ad), adeno-associated viruses (AAV) or lentiviruses (LV) [36]. A focus has been deployed towards AAV in recent years due to their interesting potential. Described in 1996, they were characterized as nonenveloped icosahedral viruses with a relatively small size (25 nm) [37] (pp. 1–23). Given these properties, AAV are favoured in gene therapy methods for the treatment of inherited retinopathies (IR); their small size enables the efficient target of retinal layers. Furthermore, AAV exhibit low immunogenicity, allowing a prolonged expression of the target gene after single-dose treatment, as well as safety second-dose administration in the subretinal space [38]. In fact, a pre-immunity towards Ad-based vectors exists due their high seroprevalence, thus compromising Ad-base gene therapies [39]. On the contrary, AAV are non-pathogenic and have not yet been linked to human diseases, even though they can be present [36].

AAV vectors are widely studied and are known for their simple genome that lacks auto-replication and expression [36]. DNA editing by AAV can be performed through the clathrin-mediated endocytosis pathway. Once in the nucleus, the DNA strand of the AAV undergoes second-strand synthesis followed by gene transcription. The AAV genome was also shown to integrate the human genome on chromosome 19 at AAVS1 position [40]. AAV are of interest due to their high transduction efficiency, their ability to edit a larger pool of cell types, their low immunogenicity, and their stable expression [41]. Major challenges regarding single AAV vector use concerns their cloning capacity being limited to non-large genomes (i.e., the cargo capacity of AAV is 4.7 kb), the nature of the procedure being highly invasive (i.e., requiring a subretinal injection), and, requiring second-strand synthesis, the onset of transgene expression can be relatively slow [38]. Scientists have sought to implement novel approaches to tackle this challenge regarding the treatment of IRD [42]. By splitting the coding sequence into separate AAV vectors, it is possible to reconstitute the full-length transgene protein expression in transduced cells [43,44,45]. This novel approach has shown to increase the cloning capacity of AAV to 14 kb when using triple vectors [46]. Furthermore, the use of dual vectors for the treatment of Stargardt disease in *Abca4*^−/−^ mice was shown to improve disease phenotype [47]. The injection of multiple vectors for the treatment of monogenic diseases comes with drawbacks; the efficiency of multiple AAV vectors is mainly compromised due to the requirement of laborious post-translational modifications such as recombination and splicing [42]. Overall, novel AAV capsids have been shown to overcome these drawbacks by allowing an intravitreal gene therapy, thus alleviating the burden induced by intraretinal injections [48,49,50].

#### 5.1.2. CRISPR-Cas Gene Editing System

The CRISPR-Cas9 gene editing system is a novel tool widely used in gene expression and suppression methods [51,52]. In retinal diseases, its use has been shown to successfully silence dominant mutations via the NHEJ (non-homologous end joining) pathway [53,54]. Although higher rates of successful gene editing have been reported in vitro, a major challenge in the clinical application of the CRISPR-Cas9 gene editing system regards its lower performance rates. The success of gene editing is highly dependent on the guide RNA (sgRNA) design, the cell target type, the delivery method (viral or non-viral vectors), and factors related to the host. Despite this, CRISPR-Cas9 gene editing may still be significant for inherited retinal diseases, as even a small percentage of gene modification can lead to an improvement in phenotype.

More recently, there has been a growing interest towards RNA-guided gene editing methods [55]. Unlike the DNA-editing CRISPR-Cas9 system, the Cas13 tool is an RNA-targeting endonuclease that can implement gene modifications during transcriptional activity [56,57]. Monogenic disease therapy using Cas13-based RNA editing has been successfully shown to restore protein product functions [55]. One of the major drawbacks associated with the Cas13 tool concerns the possibility of non-specific collateral RNA cleavage [58,59,60]. Furthermore, recent studies have shown a potential neurotoxic effect of Cas13 enzymes through the impediment of neuronal development in vitro [61]. Overall, the use of CRISPR-cas13 methods represents a novel approach for the treatment of retinal diseases, and further studies are required to assess their potential and safety [62].

#### 5.1.3. RNA Replacement

RNA replacement therapies are the process of depleting endogenous mutant RNAs. Noncoding RNA (ncRNA) are the key actors involved in gene expression regulation. Three main regulatory elements are present in ncRNA and can be classified as micro-RNA (miRNA), small interfering RNA (siRNA), and short hairpin RNA (shRNA) [63]. miRNA regulates post-transcriptional modifications by binding the binding sites of a target messenger RNA (mRNA), which further results in translational repression and mRNA destabilization [63]. The process of RNA interference (RNAi) is involved in the regulation of protein-coding gene expression [64]. Mirtrons (i.e., a subtype of miRNA) are RNAi effectors that have the potential to be used in gene therapy [65]. Their application in the treatment of RP is further discussed. siRNA, a double-stranded RNA, has a similar mechanism of action to miRNA, but its binding capacity is highly specific in comparison to miRNA being able to distinguish single nucleotide disparities [66]. shRNA, whose transcripts are similar in structure to those of miRNA, acts on DNA delivery as opposed to RNA effector molecule regulation, as siRNA does [63]. Finally, synthetic single-stranded RNAs are known as antisense oligonucleotides (ASO) and have been shown to have therapeutic effects in inherited retinal diseases [63]. They primarily exhibit their role by binding to RNAs and promoting RNA cleavage. One of the biggest challenges involved in siRNA use is its susceptibility to degradation by serological enzymes. In comparison to siRNAs, shRNAs have been shown to have a sustainable effect and higher potency [67].

### 5.2. Targeting Retinitis Pigmentosa Pathogenesis

Multiple genetic mutations have been shown, in preclinical studies, to play a role in the development of RP [68] (see Table 1). Alterations in the apoptotic and non-apoptotic cell death pathways, inflammatory responses, and metabolic pathways are amongst the mostly characterized pathways [68]. Currently, no curative treatment exists for the management of patients diagnosed with RP. To overcome these challenges, current approaches intend to develop drugs that target specific genes (e.g., gene therapies) for cell rescue, as well as molecular targets to allow cell replacement and artificial retina maintenance [69].

Multiple studies have thoroughly reviewed current treatment options in RP management, mainly focusing on those in clinical trials or those approved by the FDA [68,69]. Being a highly genetically heterogenous disease, numerous potential molecular targets are yet to be elucidated. In this section, we will review the new molecular findings (e.g., preclinical studies) involved in RP pathogenesis that can potentially lead to new drug development. Advances in mutation-specific gene therapy will also be covered.

**Table 1 pharmaceutics-15-00685-t001:** Summary of preclinical phase study advances for retinitis pigmentosa-related gene mutations. This table summarizes the new advances in preclinical phase studies that target retinitis pigmentosa-related gene mutations.

Inheritance Mode	Genes	Preclinical Phase Studies
Autosomal Dominant Retinitis Pigmentosa	*RHO*	CRISPR-cas9-induced *Rho* mutation in *Xenopus laevis* tadpoles inhibits Müller glial cell proliferation [70]. T17M rhodopsin expression induces the upregulation of IL-1β, IL-6 and NFκB and the IB1A microglia markers in C57BL6 mice [71]. Proline substitution with leucine (P347L) in rhodopsin gene in transgenic rat strain induces rapid destruction of the outer nuclear layer, by CHOP and BiP activation [72]. Subretinal injection of fused zinc-finger to the KRAB domain in combination to a *RHO* cDNA driven by the GNAT1 promotor in pigs restores *Rho* gene expression and retinal function [73]. CRISPR-cas9 gene therapy in S334ter-3 rats increases visual acuity and retinal preservation [74]. AAV-miR-2014 delivery in *RHO*-P247S transgenic mice enhances retinal function [75]
	*Nrl*	CRISPR-Cas9 engineered NRL-deficient ESC retinal organoids exhibit an abnormal number of photoreceptors expressing S-Opsin [76]. AAV-delivered CRISPR-cas9 *Nrl* gene knockdown in RHO-P347S Rd10 mice prevents retinal degeneration [77].
	*Nr2e3*	AAV-delivered CRISPR-cas9 *Nr2e3* gene knockdown in Rd10 mice prevents retinal degeneration [78].
	*PRPF*	AAV-associated CRISPR-cas9 *Prpf31* gene augmentation in mice restores retinal function [79].
	*RP1*	Dual Cas9/sgRNA successfully reduces *RP1* expression in edited Hap1-EF1a-RP1 cells [80].
Autosomal Recessive Retinitis Pigmentosa	*RHO*	HEK293 and HT1080 human cells lines expressing E249ter or W161ter mutants exhibit lower rhodopsin mRNA levels [81].
	*RPGR*	AAV-delivered CRISPR-cas9 restores RPGRORF15 reading frame in rd9 mice [82]. AAV-delivered CRISPR-cas9 restores photoreceptor preservation in *Rpgr*^−/y^Cas9^+/WT^ mice [83].
	*CNGB1*	*CNGB1* expression augmentation in *Cngb1*^−/−^ mice using an AAV vector restores vision and delays retinal degeneration [84] (pp. 733–739).
	*TULP1*	Single-knockout (tulp1^+/−^) and double-knockout zebrafish models for the expression of *Tulp1* significantly alters ciliogenesis through the downregulation of tektin2 [85]. AAV-mediated *Tulp1* gene expression editing restores *Tulp1* mRNA and protein levels [86].
	*FAM161A*	*Fam161a* knockout mice display retinal degeneration phenotype as well as enhanced molecular generative markers [87]. Homozygous *Fam161a* p.Arg512∗ have altered visual acuity due to complete loss of the outer nuclear layer and photoreceptor cell death [88].

#### 5.2.1. Physiological Signaling Pathways of the Retina

The retina of the human eye is a complex structure that involves the interplay of multiple retinal genes. Over the years, numerous gene mutations have been shown to play a crucial role in retinal dystrophy [89,90,91], some of them leading to clinical phase studies for drug development. Studies have shown that most of the genetic mutations associated with RP were due to photoreceptor or RPE dysfunctions (see Table 1) [92].

#### 5.2.2. Phototransduction

Photoreceptors (i.e., rod and cones) are highly specialized neurons that ensure the initial step of vision: phototransduction (Figure 2). Rod photoreceptors are sensitive to light and allow proper vision in dim light, whereas cone photoreceptors are sensitive to color vision. The highly compartmentalized structure is necessary for their respective functions. Each photoreceptor consists of five (5) distinct regions: the outer segment (OS), connecting cilium (CC), inner segment (IS), nuclear region, and the synaptic region. The OS of rod cells contains stacked disks, allowing to significantly increase the membrane surface area, thus promoting a high density of visual pigments (i.e., rhodopsin) [93]. Rhodopsin is a G-protein-coupled receptor synthesized in the endoplasmic reticulum (ER) that allows phototransduction once activated by light [94]. It was shown that light perception induces a conformational change following isomerization of rhodopsin to its active all-*trans* isomer form [95]. The activation of rhodopsin induces GTP-mediated signalling: GDP is exchanged for GTP on the α-subunit of the trimeric G-protein transducin, followed by the dissociation of the α-subunit-GTP heterodimer from the βγ subunits [96]. The heterodimer hence activates phosphodiesterase 6 (PDE6), leading to the hydrolysis and reduction of cGMP in the cytosol of photoreceptors [96]. Lower levels of cGMP induce the closure of the cyclic nucleotide-gate (CNG) ion channels, which results in the abolition of calcium and sodium influx, followed by the hyperpolarization of the rod photoreceptor and inhibition of subsequent glutamate release at the synapse [97].

#### 5.2.3. Photoreceptor Cell Death

Photoreceptor cell death is induced once cell homeostasis is altered. The progressive degeneration and death of rod photoreceptors is involved in visual loss and is the basis of multiple inherited retinal diseases, such as RP [99]. Apoptosis pathway activation is the most studied cell death mechanism in retinal degeneration and was shown to be involved in photoreceptor cell death (Figure 3) [100,101]. Apoptosis is a programmed cell death mechanism that involves three caspase-dependent pathways: the extrinsic, intrinsic, and ER-stress pathways [102]. Following gene mutations, the intrinsic apoptotic pathway can be activated. The intrinsic pathway is mediated by cytochrome c release; an alteration in the mitochondrial membrane potential by pro-apoptotic Bcl-2 family proteins induces pore formation and subsequent release of cytochrome *c* [102]. Once released, cytochrome c forms a complex with the apoptotic protease-activating factor 1 (APAF1) and procaspase-9 [102]. This heterotrimeric complex is called the apoptosome, which allows the activation of caspase-9. Activated caspase-9 then cleaves other executioner caspases, mainly leading to the activation of caspase-3, ultimately inducing irreversible DNA fragmentation [102]. Although apoptosis plays an important role in retinal degeneration, it was shown that inflammasome activation by NOD-like receptors (NLRs) significantly contributes to photoreceptor cell death through immunity activation and executioner caspase cleavage [103].

Studies have further demonstrated the importance of the ER stress pathway activated by the unfolded protein response signaling in retinal degeneration (Figure 3). The UPR pathway is activated once there is an upregulation of unfolded or misfolded proteins in the cytosol [104]. Three main pathways are involved in the UPR through the activation of ER transmembrane proteins: the inositol-requiring enzyme-1 alpha (IRE1α) [105], the eIF2 kinase-PERK (eIF2α-PERK), [106] and the activating transcription factor 6 (ATF6) [107] pathways. An uncontrolled activation of the ER stress pathway ultimately leads to apoptosis of cells [102]. In homeostatic conditions, this leads to the inhibitory chaperone molecule of the ER transmembrane proteins, the glucose regulatory protein 78 (GRP78/BiP) [108].

Several regulated necrosis pathways have been shown to be involved in RP pathogenesis, such as necroptosis, ferroptosis, and pyroptosis. The necroptosis pathway can be activated by TNFα or reactive oxygen species (ROS) stimulation and through ischemia-induced cell death [109]. The molecular pathways lead to the activation of RIPK1-RIPK3-MLKL, which ultimately alters membrane integrity, leading to an inflammatory response [109]. Ferroptosis is involved in lipid peroxidation and ROS level elevation through intracellular iron accumulation. Pyroptosis, similarly to the apoptosis pathway, induces caspase activation, but also induces cell membrane lysis and inflammatory response [99].

### 5.3. Autosomal Dominant-Linked Mutations

Autosomal dominant disorders are the most studied subtypes due to their severity; a single allele mutation is enough to induce a change in phenotype. Mutations involved in adRP account for 50 to 75% of cases [110]. Therefore, there is an urge to find new therapeutic approaches targeting these genes. Numerous genes are known to be involved in adRP pathogenesis. In this section, we will review the most studied gene (i.e., *RHO*), as well as the new advances regarding other molecular targets.

#### 5.3.1. Rhodopsin-Induced Autosomal Dominant Retinitis Pigmentosa

One of the most common mutations observed in patients with RP involves the *RHO* gene, which affects nearly 25% of the patients [111]. A significant effort has been deployed in preclinical studies over recent years to find a treatment for rhodopsin-associated RP. Pathogenesis and biomolecular changes induced by *RHO* mutations have been thoroughly studied in cell lines and animal models, mainly rodents. Having established robust and valid animal models for the study of rhodopsin-induced RP, scientific interest has shifted towards the development of gene therapy vectors.

Current gene therapy vector therapies under development in preclinical phases for the treatment of RP mainly target adRP, due to novel advances in mammalian models exhibiting rhodopsin mutations. Nonetheless, recently, in vivo models have allowed better characterization of *RHO* gene function in the pathogenesis of RP. Generation of a CRISPR-cas9-induced *rho* mutation in *Xenopus laevis* tadpoles has recently revealed that rod cell degeneration is due to the inhibition of Müller glial cell proliferation [70]. Proline substitution with leucine (P347L) in a transgenic rat strain showed a rapid destruction of the outer nuclear layer due to rhodopsin accumulation [72]. Further, the ER stress pathway was shown to be activated in P347L transgenic rat through significant upregulation of *CHOP* and *BiP* mRNA expression [72].

Rhodopsin mutation P347S was shown to impair the transport of rhodopsin to the photoreceptor OS in transgenic mice [112] and transgenic pigs [113]. Using transcriptional repression methods with zinc-finger DNA-binding domains, Mussolino and his colleagues have successfully transfected *RHO* P347S transgenic mice using an AAV2/8 vector containing the KRAB (Krüppel-associated box repressor) domain, which led to an increase in ERG response [114]. This method allowed to diminish transcript levels of the mutant without altering the expression of the endogenous murine *RHO* gene. Recently, it was shown that the fusion of the zinc-finger to the KRAB domain in combination to a *RHO* cDNA driven by the GNAT1 promotor could restore the wild-type copy of the RHO gene once delivered by a vector [73]. By subretinal injection, Marrocco and his colleagues have successfully suppressed the endogenous expression of the *RHO* gene containing the mutant and replaced it with a wild-type copy of *RHO* in pigs, resulting in retinal morphology and function recovery [73]. RNA replacement therapies have also shown their effectiveness in animal models. Using the P347S transgenic mice model, it was shown that shRNA delivery by a vector allowed the replacement of mutant rhodopsin RNA transcripts, which led to an elevation of rhodopsin levels and subsequent ERG response [115,116,117]. The use of the CRISPR-cas9 system has also shown its success in S334ter-3 rats, having an allele-specific disruption of *RHO^S334^*, where treatment increased retinal preservation and visual acuity [74]. Promising results were also obtained for *Nrl* [77] and *Nr2e3* [78] gene disruptions with dual subretinal injection of AAV-cas9 vector in RHO-P347S/Rd10 and Rd10/Rd1 mice, respectively. NRL and NR2E3 are crucial factors involved in rod photoreceptor cell differentiation and cell homeostasis, where their modulation was shown to be a therapeutic approach for RP [118]. Dysregulation in miRNA levels has been shown to be involved in RP pathogenesis. miR-204 mutations are involved in retinal dystrophies [119]. Karali and his colleagues have demonstrated that AAV-mediated pre-miR-204 delivery by injection in *RHO*-P247S transgenic mice enhances retinal function by delaying its degeneration and attenuating microglia activation and photoreceptor cell death [75]. Furthermore, the use of mirtron-based miRNA expression regulation has shown to successfully suppress gene expression in an RP mice model. Subretinal injection of AAV-mirtron vector (i.e., AAV-M3.M5^H^.RHO^M3/5R^) in *Nrl.GFP*/+, *Rho*^P23H/+^ mice has been shown to induced rhodopsin mRNA replacement, which partially rescued retinal degeneration [65]. Tvrm4 mice that carry a mutation of the Rhodopsin gene were also investigated to understand the effects of Myriocin, an inhibitor of ceramide de no-vo synthesis [120]. Myriocin works to lower retinal ceramides, preserve Electroretinogram Recording responses, and lower retinal oxidative stress, showing that it may act on a cell to detoxify and support cell survival [120].

Rhodopsin T17M mutants were shown to be involved in the aberrant activation of the unfolded protein response (UPR) pathway due to ER stress and induce photoreceptor cell death [71,121]. Using transgenic C57BL6 mice, it was shown that T17M rhodopsin expression induced the upregulation of IL-1β, IL-6, and NFκB and the IB1A microglia markers [71]. The persistent activation of the UPR pathway was linked to loss of photoreceptor function and retinal structure [71]. Using C57Bl/6 mice, it was shown that ATF4 single-knockdown (ATF4^+/−^) of T17M was sufficient to yield a therapeutic response by significantly delaying retinal degeneration, whereas double-knockdowns enhanced the observed effect [122]. ATF4 deficiency in T17M-induced RP mice was linked pEIF2α, ATF6, and CHOP expression attenuation [122].

#### 5.3.2. Non-Rhodopsin-Related Autosomal Dominant Mutations

Mutations in pre-mRNA processing factors (PRPFs) are linked to up to 20% of adRP cases [123]. PRPFs are involved in pre-mRNA splicing and are known to be involved in ciliogenesis, DNA damage repair pathways [123]. Using AAV-associated CRISPR-cas9 vector gene augmentation, it was shown that *Prpf31* gene augmentation restored the integrity of retinal structure in mice [79]. Similarly, in *Rpgr*^−/y^Cas9^+/WT^ mice, it was shown that *Rpgr* gene-editing using sgRNA (cas9) and AAV vectors induced photoreceptor preservation [83].

Mutations in the *RP1* gene, involved in adRP [124], disrupt protein transport within photoreceptors, cilial structure maintenance and disc membrane stability, ultimately leading to photoreceptor cell death [125,126]. Recently, dual Cas9/sgRNA was shown to successfully reduce RP1 expression in edited Hap1-EF1a-RP1 cells [80], paving the road for further clinical trials.

### 5.4. Autosomal Recessive-Linked Mutations

#### 5.4.1. Rhodopsin-Induced Autosomal Recessive Retinitis Pigmentosa

Similarly to the adRP inheritance form, numerous *RHO* mutations have been shown to be associated with the classical form of RP (e.g., adRP), whereas only a few were linked to the recessive form [127]. *RHO* mutations for adRP are classically classified into seven categories, according to phenotype induced by the mutation [127]. However, only five mutations in *RHO* have been shown to be linked to arRP [127]. The five known variants for rhodopsin-associated arRP involve the E150K, W161ter, E249ter, and M2531 mutations [127]. A missense mutation (E150K) involving RHO was shown to be present on chromosome 3 of patients from Pakistani and Turkish background with a visual impairment [128,129]. It was shown in a knock-in mice model that the E150K mutation impairs rhodopsin stabilization, which results in a progressive retinal degeneration [130]. These observations were explained by a disorganized photoreceptor disc structure which subsequently impaired normal phagocytosis [130]. Homozygous E150K mice exhibit higher Müller cell activation and higher macrophages and microglia in the retina, providing evidence of retinal immune activation [130]. Two nonsense mutations (E249ter and W161ter) were found to be involved in arRP. Using HEK293 and HT1080 human cells lines transfected with the rhodopsin nonsense mutants, lower levels of rhodopsin mRNA were detected, suggestive of a nonsense-mediated decay (NMD); treating cells with Wortmannin, a known inhibitor of NMD, restored rhodopsin mRNA levels [81]. M2531 rhodopsin mutants were shown to be linked to the arRP form, where heterozygous patients from consanguineous Turkish descents were asymptomatic [131]. Further studies are required to better understand the functions and impacts of autosomal recessive mutations in the development of RP.

#### 5.4.2. Non-Rhodopsin-Related Autosomal Recessive Mutations

*CNGB1* sequence variants were shown to be associated with arRP, accounting for nearly 4% of arRP cases [132]. A recent literature review on *CNGB1* variant has shown that a total of 62 genetic variants were linked to inherited retinal diseases [132]. *CNGB1* expression restoration in *Cngb1*^−/−^ mice using an AAV vector was shown to restore vision; *CNGB1*-expressing mice had a higher performance in a rod-dependent vision-guided behavior test [84] (pp.733–739). Furthermore, retinal degeneration was shown to be significantly delayed, with morphology preserved [84] (pp.733–739).

*TULP1* mutation causes early-onset retinal degeneration, although its pathogenesis is yet to be fully discovered [133]. The use of the zebrafish model has gained interest in the past years to study the pathogenic mechanism of various retinal diseases [134]. Single-knockout (*Tulp1^+/−^*) and double-knockout zebrafish models for the expression of *Tulp1* were generated and significantly altered ciliogenesis through the downregulation of tektin2, a microtubular component [85]. Using AAV-mediated *TULP1* gene expression editing, *TULP1* mRNA and protein levels were successfully restored [86]. However, *TULP1* gene expression restoration is not enough; in *Tulp1^−/−^* mice, the gene editing process did not improve the thickness of the outer nuclear layer [86].

Establishing new animal models for gene therapy development is one of the leading solutions for drug discovery. Recently, Beryozkin and his colleagues have successfully generated a knockout mice model for *Fam161a* deficiency; *Fam161a* knockout mice display retinal degeneration phenotype as well as enhanced molecular generative markers such as microglia [87]. Similarly, homozygous *Fam161a* p.Arg512∗ were shown to have altered visual acuity due to complete loss of the outer nuclear layer and photoreceptor cell death [88].

The retinitis pigmentosa GTPase regulator (RPGR) gene has been shown to induce recessive RP and is the most commonly studied gene [135]. Although there are promising clinical trials in AAV vectors for therapeutic approaches, preclinical phase studies are still being conducted for the development of miscellaneous gene therapy methods. Recently, using the rd9 mice model, the AAV-guided CRISPR-cas9 vector injection restored the reading frame of *RPGR^ORF15^* [82].

### 5.5. Identification of Gene Targets Involved in Retinitis Pigmentosa for Novel Gene Therapy Treatments

There is an increasing demand for the identification of novel genes involved in the pathogenesis of RP for future gene therapy development. Gene sequencing studies using patients from large cohorts shed light on future gene targets. Numerous studies in the past five years have demonstrated a potential involvement of multiple genes in RP pathogenesis through genome sequencing [136,137,138]. For example, analyzing sequence variants of the solute carrier (SLC) genes in Israeli and Palestinian inherited retinal disease cases, two novel targets were identified: SLC66A1 and SLC39A12 [139]. Other variants of SLC genes were shown to be involved in RP; SLC7A14 knockout mice show retinal degeneration and altered visual function, suggesting a role for this gene in retinal development [140]. Rare heterozygous variants of the ATP/GTP binding protein like 5 (AGBL5), p.Arg281Cys and p.Arg487*, were also shown to be potentially involved in RP [141]. Overall, the pathophysiology of these novel genes in RP remains to be elicited. An updated list of genes and mapped loci causing retinal diseases can be found on the Retinal information Network’s website (https://web.sph.uth.edu/RetNet/ (accessed on 20 January 2023)).

## 6. Stem Cell Therapy

Other forms of vision restoration include cell therapy. Cell-based therapy works in two forms: (1) to replace dysfunctional cells with effective stem cells, and (2) for the restoration of the dysfunctional cells by releasing trophic factors. For transplanted cells to be functional, they must integrate in the long term and form new synaptic connections with the host [142].

### 6.1. Various Models and Sources

#### 6.1.1. Embryonic and Pluripotent Stem Cells

Embryonic stem cells (ESCs) are pluripotent stem cells that can self-renew through division and develop into the three primary germ cells. A drawback of using ESCs is their potential to be rejected by the immune system. As an alternative, induced pluripotent stem cells (iPSCs) are used. iPSCs are generated from a somatic cell line and can differentiate into any somatic cell. This generation of iPSCs allows for stem cell extraction without using human embryos. The dysfunction of the retinal pigment epithelium affects the photoreceptors and causes significant vision loss. Replacing the damaged retinal pigment epithelium and photoreceptors with healthy pluripotent stem cells can delay the disease progression and potentially restore vision loss [143,144].

Since 2004, many reports have noted the potential of iPSCs implantation to improve vision in rat models [145]. In 2021, Surendran and his colleagues investigated human induced pluripotent stem cell (hiPSC)-derived retinal cells in mice. Their results showed improvements in the functions of photoreceptor progenitor (PRP) cells and the retinal pigment epithelium (RPE) through the expression of cone and rod markers, neural retina proteins, and KCL- induced polarization [146]. hiPSC therapy’s effect could further be explained by increased pigmentation and ciliation, enhanced tight junction protein expression and secretion of various growth factors [146]. Other studies have shown ESCs differentiated into retinal organoids, and their transplant into rats exhibited significant improvement in visual function. The transplanted retinal organoid survived for seven months and showed good integration into the host [147]. Recently, hiPSC-derived RPE cells injected into an atrophic retina of a swine model showed a typical epithelial morphology, expressed RPE-related genes, and had phagocytic ability [148]. These results show that hiPSC-RPE can slow down progression of disease [148]. Moreover, the transplantation of combined RPE sheets with retinal sheets in the subretinal segment of RSC rats showed their potential in complete replacement of degenerated retina; this therapy could be used in advanced stages of retinal degeneration [149].

The utilization of iPSCs can remove the ethical concern regarding the use of human embryos and can match patients based on compatible blood types. However, some drawbacks include epigenetic memory, where the derived cells retain gene expression from the original cells, potentially impacting properties like senescence and proliferation. Additionally, the ability of iPSCs to proliferate indefinitely raises significant safety concerns, such as the formation of teratomas.

#### 6.1.2. Bone Marrow Stem Cells Therapies

Bone marrow stem cells (BMSCs) are in the bone marrow and include two types: (1) mesenchymal stem cells (MSCs) and (2) hematopoietic stem cells (also referred to as CD34^+^ cells). BMSCs are multipotent cells that can generate cells specific to a particular tissue; however, their differentiation capacity is more limited than pluripotent cells. BMSCs are mainly looked at when studying RP [150]. The ability of BMSCs to fully differentiate into photoreceptors continues to be under investigation, but preclinical studies have demonstrated that BMSCs release anti-angiogenic and neurotrophic factors, as well as immunomodulatory proteins such as insulin-like growth factor-1 (IGF-1), class II major histocompatibility complex (MHC class II) antigens, and Th2-related cytokines [151,152,153]. Brown and colleagues used primitive MSC-derived retinal progenitor cells in rd12 mouse models. They tracked PKH26 dyed cells and found that cells were able to integrate and counter inflammation [154]. Overall, the advantages of BMSCs include their ability to migrate towards lesion sites, the ability for trans-differentiation, and the simplicity in extracting and manipulating CD34+ cells.

#### 6.1.3. Therapies Based on Stem Cell-Derived RPE

Stem cell-derived RPE can be used as supportive cells to provide trophic support for surviving photoreceptors [155]. Early phase clinical trials with encapsulated RPE cells producing ciliary neurotrophic factors suggested photoreceptor protection in patients with RD [156]. Moisseiev and colleagues administered CD34^+^ cells into C3H/HeJ mice, which are homozygous for the rd1 mutation, and found molecular changes which suggested trophic regenerative effects, but with no changes in electroretinography (ERG) [157]. Recently, conjunctiva mesenchymal stem cells have been induced into photoreceptor-like cells in fibrin gel to form 3D scaffolds [158]. The application of bioengineering of cells in this manner is a promising strategy to increase the number of photoreceptor cells and promote the angiogenesis needed in the retina for repair and regeneration [158]. In vivo studies on mice have revealed that transplanting mesenchymal stem cells (MSCs) into the vitreous can improve the survival and preservation of photoreceptors, thereby delaying the progression of retinal degeneration in retinitis pigmentosa (RP) [159]. This treatment inhibited the NF-κB pathway and upregulated anti-inflammatory cytokines, while downregulating pro-inflammatory cytokines [159]. Additionally, transplanting both fetal retinal pigment epithelium (RPE) cells and MSCs in mice has been found to significantly improve ERG results and increase the survival rate of transplanted cells through an increase in the expression of Crx, a protein involved in activating rhodopsin and rhodopsin levels, as well as a decrease in caspase-3 expression [160]. These results suggest that there is greater benefit in coculture transplantation compared to a single-cell transplantation [160].

#### 6.1.4. Therapies Based on Retinal Progenitor Cells

Retinal progenitor cells (RPCs) are mitotically active multipotent stem cells found in the developing neural retina and obtained from human fetuses between 16 to 20 weeks of gestation [161]. These immature cells can be manipulated in vitro to express photoreceptor markers and differentiate into neuronal cells of the retina [162,163]. Preclinical studies have shown that retinal precursor cells at the peak of rod genesis can differentiate into rod photoreceptors when transplanted and integrate into the degenerating retina to form synaptic connections and improve visual function [164]. Calcium imaging studies on mice transplanted with C-Kit+/SSEA1− (C-Kit+) RPCs derived from organoids have revealed exceptional results in the formation of functional synapses with host cells, potentially stabilizing progressive vision loss [165]. Overall, the advantages of using RPCs include the secretion of trophic factors to increase the likelihood of retinal survival and photoreceptor replacement [166].

In short, while clinical trials are yet to be conducted, preclinical studies have shown promising potential for stem cells to regenerate the retina. The ability to differentiate retinal progenitor cells into photoreceptors and the integration of transplanted cells into the degenerating retina, along with the formation of functional synapses between host cells and transplanted cells, all indicate that stem cell therapy has the potential to stabilize or reverse progressive vision loss.

## 7. Novel Therapeutic Targets in Preclinical Phase: Optogenetics

While retinal gene therapies have had significant successes over the years, there are cases in which they cannot compensate for the loss of function a patient has. Some limitations of gene therapy include patients without genetic diagnosis and patients with advanced diseases with no remaining photoreceptors [167]. In turn, optogenetics is a one-for-all therapy that can be used in cases where degradation has occurred, regardless of the specific mutation present. In RP, the basic idea of optogenetics is to convert non-light-sensitive retinal cells, typically bipolar or retinal ganglion cells (RGCs), into artificial photoreceptors. This is achieved by introducing a light-sensitive protein (i.e., opsin) into the cells. Since the discovery of channelrhodopsin-2 (ChR2), optogenetic tools have been explored to make cells light-sensitive [168]. Depending on the methods and target cell, depolarized and/or hyperpolarized opsins can be used. Depolarized opsins are employed to mimic an “on” response, whereas depolarized opsins are utilized on dormant cells [169]. Two superfamilies—microbial opsins (Type 1) and animal opsins (Type 2)—are utilized to categorize the opsin genes employed in optogenetic vision restoration. While both opsin families contain photoactive proteins with seven transmembrane α-helical domains, their light sensitivity, function, and utility for vision restoration differs [170]. Type 1 opsins use the all-trans-retinal chromophore and isomerize upon light absorption to cause a conformational shift and directly affect ion channels or pumps [171]. Type 2 opsins often attach covalently to 11-cis-retinal, and light absorption causes intracellular G-protein-coupled receptor (GPCR) signaling cascades, which in turn affect ion channels indirectly [172].

### 7.1. Microbial-Derived Opsins

Garita-Hernandez and colleagues sought to understand the membrane trafficking efficiency and toxicity of microbial opsin from human retinal organoids derived from hiPSCs [127]. Depolarizing and hyperpolarizing microbial opsins including CatCh, ChrimsonR, ReaChR, eNpHR 3.0, and Jaws were tested. The highest membrane localization was found in eNpHR 3.0, ReaChR, and Jaws, explained by an increased release of ER and membrane trafficking signals [127]. Membrane localization was less efficient in CatChR and ChrimsonR due to protein accumulation in the cytosol at high doses, ultimately leading to UPR [173,174]. The results of this study suggested hiPSC-derived retinal organoids are predictive of optogenetic proteins in human retinal contexts, as well as providing information about expression patterns of microbial opsins and their effect on health for future studies [175]. Recent advances in common optogenetic approaches demonstrated that optogenetic therapy allows cell restoration and vision improvements through the transplantation of wild-type donor-derived photoreceptor precursors using HEK293 triple-transfected cells and/or hiPSC-derived photoreceptors equipped with microbia opsin into *Cpfl1/Rho*^−/−^ mice and wild-type mice (C57BL/6J) [175].

### 7.2. Animal-Derived Opsins

Ganglion cells are the targets of many large animal studies. Through intravitreal injection, the delivery of modified mammalian ionotropic glutamate receptor (LiGluR) to the ganglion cells of an *Rcd1* dog showed restored vision in visible light [176]. This acquired properties of light-gated channels upon binding with maleimide-azobenzene-glutamate-0, showing evidence of derived light responses [177]. Similar delivery of optogenetic therapy targeting the outer retina was performed in *Rdc1* dogs and C57BL/6 mice to investigate the differences in treatment on larger animals with higher life expectancy, showing a partial improvement in vision restoration through Y-maze performance and electroretinography [178].

Promoters have also been shown to play a crucial component in optogenetics. A ubiquitous promoter, cytomegalovirus, was used in combination with an AAV2 vector to improve upregulation of ChR-Ca^2+^-permeable ChR and activate retinas of macaques to improve light sensitivity. CatCh expression increased due to the promoter region of the gamma-synuclein gene, which is abundantly expressed in ganglionic cells [179]. The optimization of ChRs light sensitivity through two CoChR mutants (i.e., CoChR-L112C and CoChR-H94E/L112C/K264T) delivered with an AAV2 vector has shown to restore light sensitivity in TKO mice [180]. Chloromonas oogama, a specie of green algae, is a known variant described to improve light sensitivity in blind mice and restore vision under ambient light conditions [180].

It is important to note that continuous light exposure and UV light are a concern as they induce the death of photoreceptor cells. To understand how optogenetic genes respond to constant light, Tabata and colleagues exposed mice transduced with optogenetic gene *mVChR1* (using an AAV vector) to continuous light for one week and showed no reduction in visually evoked potentials (VEPs) amplitudes in RGCs, as well as an absence of phototoxicity by continuous light exposure [181]. Following this study, light sensitivity was tested under three mutations of opsins by replacing amino acids related to ion-conducting pathways in mVChR1 with corresponding ones from ChR2. The third extracellular loop in mVChR1 was found to be implicated in sodium ion selectivity, and transmembrane 6 and C-Terminal regions have a role in some channel kinetics [182]. G-protein simulation has also been shown to restore light sensitivity [183]. Gloeobacter and human chimeric rhodopsin (GHCR) under the control of the hybrid promoter comprising the CMV immediate-early enhancer, CBA promoter, and CBA intron 1/exon 1 was injected with AAV2 into mice [184]. Overall, the results of these studies investigating different variants and promoters can provide patients with more options for treatments based on their specific needs. They also identify the need for further testing regarding amino acid sequences that determine wavelength sensitivity.

In summary, studies in animals have shown that optogenetics can partially restore vision in models of RP. Nevertheless, this technique remains in preclinical stage, and more research is needed to fully understand its safety and efficacy in humans. In addition, the optimization of optogenetics using different variants of opsins and promoters might provide a better outcome for patients with RP.

## 8. Novel Therapeutic Targets in Preclinical Phase: Neuroprotective Agents

In the early stages of Retinitis Pigmentosa (RP), a combination of various treatments, including neuroprotective agent therapy, can slow the progression of the disease. Neuroprotective agents, such as antioxidants, anti-apoptotic agents, and neurotropic factors, including ciliary neurotrophic factors (CNTF) [185,186], brain-derived neurotrophic factors [186], and fibroblasts growth factors [187], can be administered as a preventative measure to protect the nervous system from damage (Figure 4). As current knowledge explains, neuroprotective agents work by inhibiting apoptosis and inflammatory processes and by reducing oxidative stress and free radicals [188]. To date, CNTF is mainly used to slow retinal degeneration, for which it has been clinically proven [189,190,191].

### 8.1. Neuroprotective Pathways

The signaling pathways of transforming growth factor (TGF)-β signaling, G-protein activated signaling, and vascular endothelial growth factor (VEGF)-mediated signaling in mice carrying the transgene rhodopsin V20G/P23H/P27L (VPP) [192] have been investigated. The results showed a strong cluster of genes regulating the VEGF, TGF-β, and G-protein-activated signaling pathways.

As it relates to the TGF-G pathway, Tgfbr2 was significantly unregulated in VPP retinae, specifically in the INL and ONL. Tgfbr2 was also observed in resting and reactive Müller cells, indicating their relationship with neuronal and glial cells [192]. RNAseq analysis also revealed an increase in the expression of genes involved in the G-protein-activated signaling family. Focusing on the upregulation of Edn2 and Ednrb after photoreceptor damage, the RNAseq data showed a significant increase in Ednrb expression in the VPP retinae [192]. Edn2 has been a known protector against photoreceptor degeneration [193]. Vegfr2 was seen to be significantly upregulated in VPP retinae and the neuronal layers of the retina. In control retinae, Vegfr2 mRNA in situ hybridization revealed multiple signals in the inner nuclear layer that partially overlapped with resting Müller cells that were positive for glutamine synthetase [192]. Vegfr1 activation has also been seen to provide neuroprotection and to be involved in RGC survival [194].

### 8.2. Neurotropic Factors

However, they have a relatively short lifespan before they degrade [195]. Yang and colleagues showed significant survival time of retinal ganglion progenitor cells (RGPCs) when CNTF and OSM were integrated into a stable and biocompatible polysaccharide nanoparticle (Nps). This was seen in vitro with RGPCs and photoreceptor progenitor cells (PPC) and in vivo with intravitreal delivery in rats [196]. This study further shows CNTF’s ability to upregulate protease inhibitors [196].

### 8.3. Anti-Apoptotic Agents

Taurine deoxycholic acid (TUDCA) is an anti-apoptotic agent with chemical chaperone activity that improves protein-folding capacity [197]. TUDCA intraperitoneal injection in *Rpgr* KO mice has shown to decrease photoreceptor cell death and inhibit retinal inflammation [198]. It is hypothesized that TUDCA reduces apoptosis through caspase-3 regulation [199]. Zhang and colleagues confirmed the effect of TUDCA in *Lrat*^−/−^ mice by demonstrating a significant loss of caspase-3 activation with subsequent reduction in apoptosis [200]. The mechanism of action of TUDCA could partially be explained by IL-1β signaling inhibition [200]. More recently, TUDCA injected into homozygous P23H line-3 rats exhibited a decrease in photoreceptor degeneration and partially protected vascular damage and glial activation [197]. Results also showed an increasing number of astrocytes which, based on current literature, provides inconclusive results about whether they are neurotoxic or neuroprotective [201]. Metformin was also found to have a neuroprotective effect on the retina in N-Ethyl-N-nitrosourea induced rat model of RP, through the upregulation of caspase-3, iNOS, CD68 (macrophage marker), and glial fibrillary acidic protein expression [202]. The effects of Cav1.3 L-type channels on neuroprotection have also been seen to provide short-term protection in rd10 mouse models. This was seen through the reduction of Ca2+ influx in an attempt to slow down apoptosis [203].

Nerve growth factor (NGF) is another type of neuroprotective agent that acts on the central and peripheral neurons and the visual system [204,205]. Treatment of PC12 cell line with NGF eye drops has shown to downregulate Bax expression and increase Bcl-2 expression [206]. Previous studies indicate this is a positive effect of NGF treatment and increases neuritis outgrowth and rhodopsin expression [207]. Overexpression of IL-21 also leads to the possible activation of Müller cells; however, this idea needs further investigation [208] (pp. 255–263). Recently, NGF was tested in zebrafish to find that intravitreal administration of rhNGF potentially has a regenerative effect on photoreceptors cells. Zebrafish were studied for 21 days after administration, and ERK1/2 pathways showed an upregulation compared to controls during early timepoints, which led to improved outer nuclear layer (ONL) thickness [209]. Similarly, the significantly increased levels of prototype galectin-1 (Drgal 1-L2) also play a role in retinal development and photoreceptor regeneration [209]. Transforming growth factor β (TGFβ) was injected in Müller cells as it is thought to have neuroprotective properties; mice with cell type-specific deletion of *Tgfbr2* in retinal neurons and Müller cells (Tgfbr2ΔOC), in combination with a genetic model of photoreceptor degeneration (VPP), were used to analyze TGFβ signaling [210]. While retinal morphology was not altered from TGFβ signaling, the upregulation of the MAP kinase pathway and the activation of pro-apoptotic genes markedly accelerated VPP-induced photoreceptor degradation in mice (Tgfbr2OC; VPP) [210]. Due to this, TGFβ signaling in Müller cells and retinal neurons likely has a neuroprotective effect and may present novel therapeutic opportunities to halt photoreceptor deterioration [210]. Similarly, it was shown that Müller cells have a vital role in support, metabolism, and modulation of neuronal excitability through transporting and releasing neurotransmitters [211].

Furthermore, *Egr1* mRNA was found to be heavily upregulated in early degeneration of rods and cones in r10 mouse models. *EGR*1 was also detected in Müller glia, suggesting EGR1 could be involved in early stress response [212]. The rap guanine exchange factor 4 (EPAC2), activated by cyclic AMP in rd10 models, also exhibited neuroprotective capacities [213]. Recently, it was similarly noted that P23H rhodopsin knock-in mice treated with 4-phenylbutyric acid (PBA) regulated stress markers, and its properties as a neuroprotector should be further explored [214].

### 8.4. Antioxidant Agents

Oxidative stress has been suggested to promote the degradation of rods and cons in RP [215]. The combination of four antioxidants (vitamin E, a SOD mimetic, vitamin C, and a-lipoic acid) was evaluated in rd1 mice. Two biomarkers of oxidative damage, carbonyl adducts measured by ELISA and immunohistochemical staining for acrolein, were found to be upregulated in the retina [216]. Komeima and colleagues also noted that the four antioxidants significantly decreased the buildup of oxidized lipids and shielded cone cells from degeneration [216]. N-acetylcysteine (NAC), which restores intracellular GSH when administered orally to rd10 mice, was shown to significantly shield rod and cone photoreceptor cells [217]. Additional neuroprotective and immunosuppressive effects of oral NAC on rd10 mice were also noted [218].

The difficulty with antioxidation when it comes to neuroprotection is the dosage of the drug, and most of these studies are performed at early stages of the disease. Additionally, many treatments using neuroprotective agents have been tested on mice as the therapy is well-tolerated and has few side effects. Some notable treatments include docosahexaenoic acid (DHA) [219,220], vitamin A [221], various retinoids and their derivatives [222], calcium channel blockers [223], calpain inhibitors [224], and valproic acid [225].

To sum up, neuroprotective agent therapy has the potential to slow the progression of RP in early stages of the disease. The use of neuroprotective agents in combination with other treatments such as optogenetics, stem cell therapy, and gene therapy may provide a better outcome for patients with RP.

## 9. Novel Therapeutic Targets in Preclinical Phase: Exosomes

Exosomes, small vesicles released by cells, have recently garnered attention as a potential therapeutic option for RP due to their ability to deliver biologically active molecules to specific cells and tissues. In this section, we will discuss the current state of exosome research in the treatment of RP and the potential mechanisms by which exosomes may be able to protect photoreceptor cells.

Exosomes are a type of extracellular vesicles that are released by cells and are found in all biological fluids. They play a role in a variety of physiological and pathological processes, including immune response, cardiovascular diseases, central nervous system diseases, and cancer. These nanosized structures (30–100 nm in diameter) contain a variety of molecules such as lipids, nucleic acids, and proteins (i.e., signaling proteins, metabolic enzymes, and antigens) which they acquire when they are released from the parent cell by budding. This also means that they take on some of the characteristics of the parent cell’s membrane.

Having shown that damaged RPE releases exosomes to control non-damaged cells and promote angiogenesis [226], several preclinical studies have leaned on investigating the potential therapeutic effects of exosome (Table 2). ARPE-19 cell line (a widely used model for RPE) stimulation with inflammatory cytokine induces the release of extracellular vesicles [227]. It was shown that ARPE-derived exosomes (RPE-Exos) inhibit T-cell proliferation through TNF-α and IL-6 secretion [227]. The promising results obtained in vitro gave rise to the study of exosome treatment in animal models. In fact, MSC-derived exosomes (MSC-Exos) were shown to induce positive effects in the treatment of RP. Using Sprague–Dawley rat models with hyaluronic acid-induced retinal detachment [228] and the Rd10 mice model [229], it was shown that the subretinal injection of MSC-Exos significantly lowers levels of pro-inflammatory cytokines (e.g., TNFα and ILβ) and suppresses photoreceptor cell apoptosis. MSC-Exos-treated Rd10 mice also displayed suppressed activation of the immune response through inhibition of microglia, Müller cells, and macrophages, as well as an inhibition of the NFκB pathway [159]. It was hypothesized that the mechanism by which exosomes exerts their effects is linked to the presence in the small vesicles of anti-inflammatory proteins, as well as markers which have a neuroprotective and anti-apoptotic effect [228]. More recently, further investigations demonstrated that MSC-Exos increase survival of photoreceptor cells and subsequent visual function improvement through transcriptional regulation [159]. Following genetic screening and the identification of possible downstream targets of MSC-Exos, it was shown that an upregulation of miR-146a in co-cultured 661W and BV2 cells treated with MSC-Exos decreases the expression of the transcription factor Nr4a3 [159]. NR4a3 is a member of the orphan nuclear receptor NR4A family and is known to be involved in peripheral T-cell responses, thus promoting inflammation [230]. Overexpressing miR-146a can significantly decrease the expression of LPS-induced pro-inflammatory cytokines [159], rendering its use clinically significant in the treatment of RP. RPE-derived exosomes (RPE-Exos) have also shown promising results in the treatment of RP. Recently, in C57BL6 mice with N-methyl-N-nitrosourea (MNU)-induced retinal detachment, it was shown that RPE-Exo treatment restores visual function by rescuing photoreceptor cells, significantly lowering TNFα and ILβ mRNA expression and inhibiting apoptosis pathway activation [231]. Similarly, MSC-Exos administration was shown to inhibit apoptosis induced by MNU and block retinal degeneration through the upregulation of miR-21 involved in key regulatory pathways [232,233]. Grafting human neural progenitor cells (NPC) in Royal College of Surgeons (RCS) rats is linked to exosome secretion [234]. Administration of mouse NPC-Exos was shown to decrease photoreceptor cell apoptosis in RCS rats through retinal microglia activation inhibition, as well as TNFα, ILβ, and COX-2 inhibition [234].

The mechanisms by which exosomes may be able to protect photoreceptor cells in individuals with RP are not fully understood. However, it is thought that exosomes may act through several mechanisms, including the transfer of growth factors and other biologically active molecules to recipient cells. This leads to changes in gene expression and cellular function, modulation of immune responses, and suppression of apoptosis [235]. Additionally, numerous advantages are in the favor of exosome use for retinal disease treatment (Figure 4). One of the biggest challenges in ocular drug delivery by systemic administration is the impossibility to cross blood-retinal and blood-aqueous barriers due to the size of the protein carrier [235]. The use of exosomes may leverage drug delivery challenges; exosomes are able to bypass these barriers due to their small size, allowing them to directly target photoreceptor cells in the retina [236]. Moreover, exosomes can selectively act on specific tissues and cells due to their ability to express different types of surface molecules [237]. They can also remain in the ocular structure for a long time due to their bilipid membrane, giving them stability and structural rigidity and preventing them from enzymatic degradation [238]. Furthermore, having shown that exosomes are naturally present in the body fluids [239], they display higher stability in physiological conditions, as well as in an inflammatory microenvironment, rendering them more biocompatible [235]. In comparison to MSC therapy, exosome-based therapies have been shown to lower teratoma formation and embolization risk, two major possible side effects of MSC therapy. Intravitreal injection of AAV2-Exos in mice has been shown to have greater penetrability in the retina [240].

Despite the promising results of these studies, there are several challenges to the development of exosomes as a therapeutic option for RP [235]. One challenge is the need to optimize the isolation and purification of exosomes; currently, isolation procedures have a low productivity rate. Additionally, there is a need for more robust preclinical and clinical data to demonstrate the safety and effectiveness of exosomes in the treatment of RP. Many key components of exosomes are yet to be elicited. Finally, there is a need to understand the long-term effects of exosome therapy and to identify the optimal dosing regimen.

## 10. Conclusions

Retinitis pigmentosa (RP) is a genetic disorder affecting the retina and leading to progressive vision loss. The classification, epidemiology, clinical manifestations, and prognosis of RP have been briefly discussed in this review. Conventional treatments for RP, such as vitamin A supplements, protection from sunlight, visual aids, and surgical interventions, have helped to slow the progression and alleviate the symptoms of RP, but they do not address the underlying genetic cause of the disease.

Recent therapeutic advances, such as gene therapies like voretigene neparvovec (Luxterna) for RPE65, have brought new hope for people living with RP. These therapies aim to target the underlying genetic cause of the disease and offer the potential to stop the progression of RP, offering a cure for some patients. However, currently, only a small number of patients with the RPE65 mutation can benefit from this treatment. Most other gene therapy targets are in preclinical phase of research and require further study prior to clinical translation.

Other than gene therapy, other preclinical therapeutic modalities on the horizon include stem cell therapy, optogenetics, neuroprotective agents, and exosomes. These have shown promising results in in vitro and animal models and offer the hope of reversing the damage, restoring vision, and revolutionizing the way RP is treated.

Retinitis pigmentosa may be a difficult condition to understand and treat. With advances in our understanding of its molecular mechanisms, we can unlock new doors to innovative therapies that will change the way we approach RP and offer a ray of hope for those suffering from this disease.

## Figures and Tables

**Figure 1 pharmaceutics-15-00685-f001:**
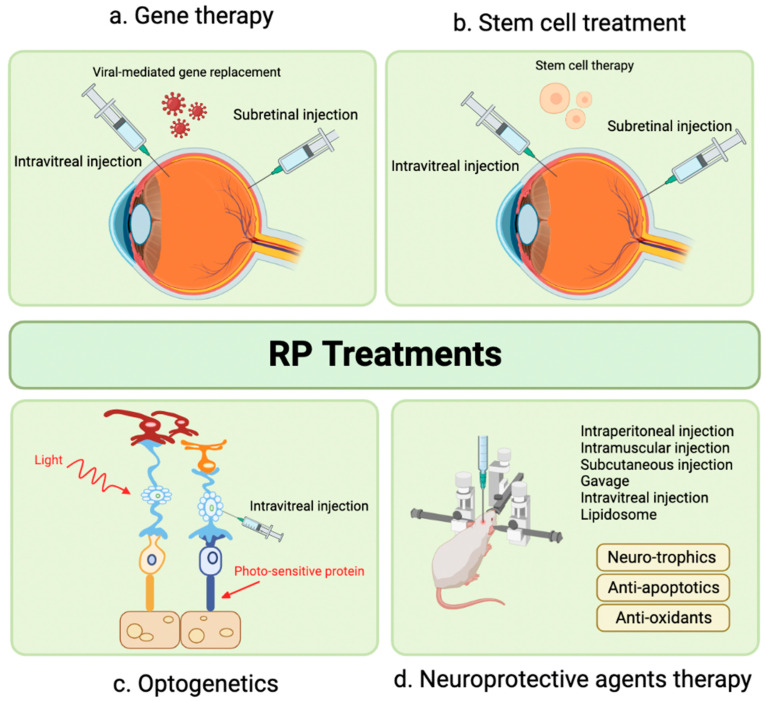
Advanced therapeutic modalities for retinitis pigmentosa. (**a**) Gene therapy, using virus-mediated injection of therapeutic gene to replace disease-causing gene; (**b**) Cell therapy, by injecting stem cells to replace injured cells and form synaptic connections with remaining retinal neurons; (**c**) Optogenetics, by introducing photosensitive proteins into degenerated retina to restore cone function, and (**d**) Neuroprotective agents, such as neurotrophic factors, anti-apoptotic agents and antioxidants, used in early stages of the disease and as adjunctive treatment. Adapted by BioRender.com (2023). Retrieved from https://app.biorender.com/biorender-templates (accessed on 20 January 2023).

**Figure 2 pharmaceutics-15-00685-f002:**
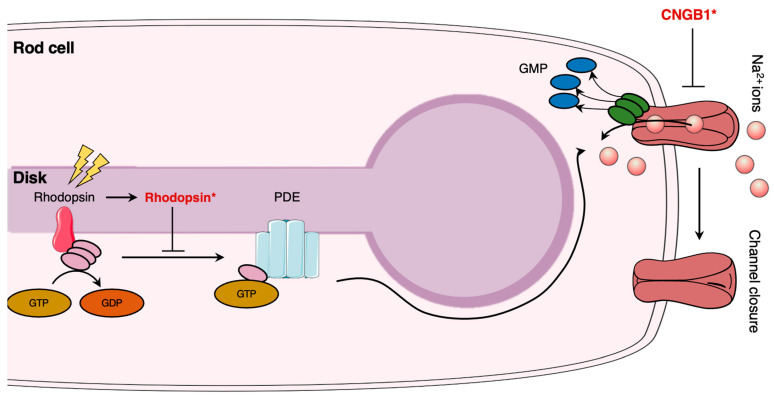
Schematic illustration of phototransduction mechanism and impacts of retinitis pigmentosa pathogenesis. In rod cells, light exposure induces transducer activation and subsequent phosphodiesterase (PDE) expression. PDE induces sodium channel closure. Rhodopsin (Rhodopsin*) and cyclic nucleotide-gate (CNG) ion channel mutations (CNGB*) alter rhodopsin signalization and are involved in retinitis pigmentosa pathogenesis. The figure was adapted from [98]. The Figure was partly generated using Servier Medical Art, provided by Servier, licensed under a Creative Commons Attribution 3.0 unported license.

**Figure 3 pharmaceutics-15-00685-f003:**
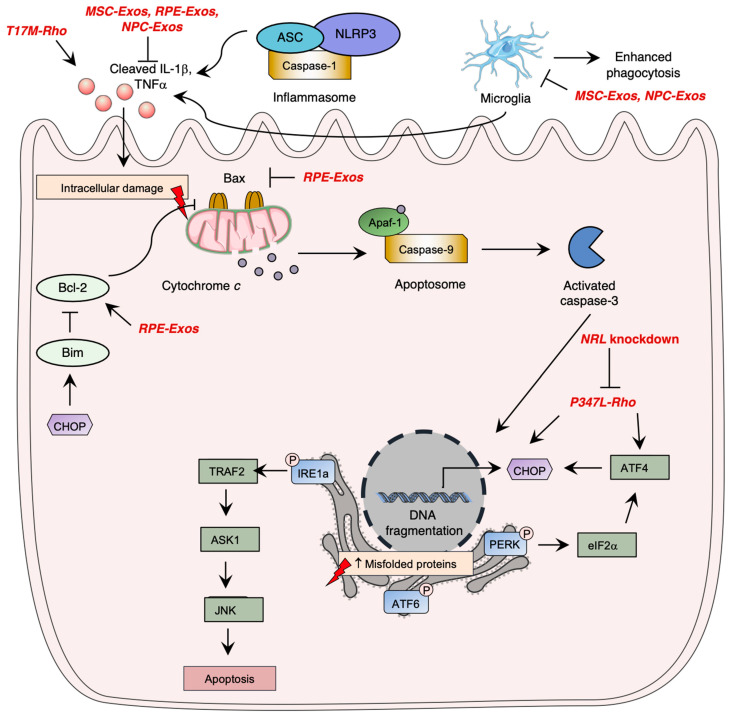
Overview of photoreceptor cell death pathways and their alteration in retinitis pigmentosa. Photoreceptor cell death is mainly mediated by (1) the activation of the intrinsic apoptosis pathway through cytochrome c release, (2) endoplasmic reticulum (ER) stress pathway through the activation of IRE1a, PERK, or ATF6, and (3) through inflammasome activation in macrophages. Mutations in XX have shown to induced early photoreceptor cell death in retinitis pigmentosa, leading to vision loss. Rhodopsin mutations enhance CHOP expression. The Figure was partly generated using Servier Medical Art, provided by Servier, licensed under a Creative Commons Attribution 3.0 unported license.

**Figure 4 pharmaceutics-15-00685-f004:**
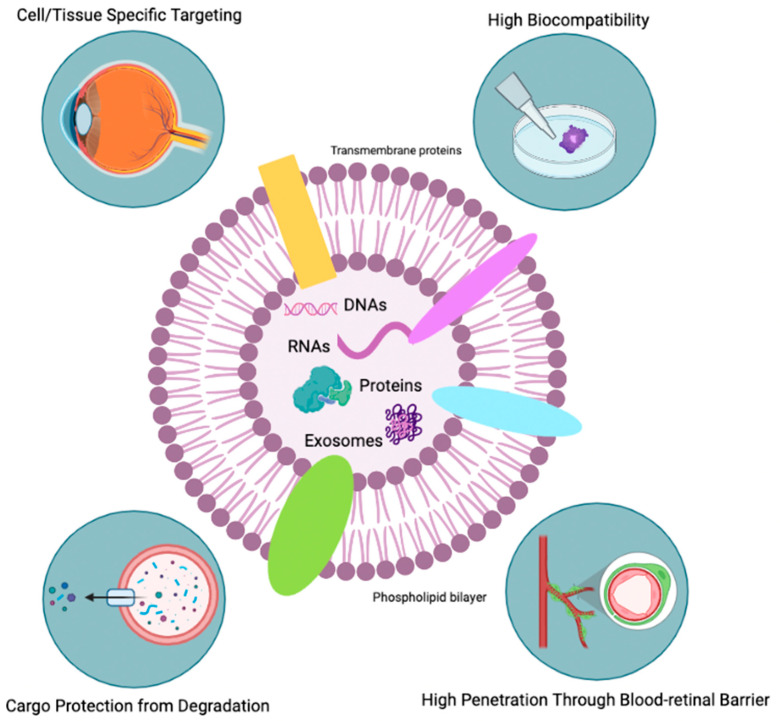
Favorable characteristics of exosomes. This Figure illustrates the favorable characteristics of exosome therapy, including its ability to target specific cells and tissues, high biocompatibility, resistance to degradation, and capacity to penetrate barriers such as the blood-retinal barrier. Adapted by BioRender.com (2023). Retrieved from https://app.biorender.com/biorender-templates (accessed on 20 January 2023).

**Table 2 pharmaceutics-15-00685-t002:** Summary of preclinical studies involving exosome use for the treatment of retinitis pigmentosa. This table provides a summary of current preclinical studies investigating the use of exosomes in the potential treatment of retinitis pigmentosa. We cover functions of mesenchymal stem cell-derived exosomes (MSC-Exos), retinal pigmented epithelium-derived exosomes (RPE-Exos), and neural progenitor cell-derived exosomes (NPC-Exos) on different animal models.

Exosome Source	Models	Observed Effect
MSC-Exos	Hyaluronic acid-induced retinal detachment in Sprague–Dawley rat	Diminished TNFα and IL-β expression Enhancement of LC3-II and LC3-I ratio Decrease of Atg5 cleavage Decrease in photoreceptor cell apoptosis [228]
	Rd10 mice	Diminished TNFα, IL-β and IL-6 expression Enhanced photoreceptor cell survival Microglial, Müller cell and macrophage activation inhibition miR-146a upregulation [159,229]
	N-methyl-N-nitrosourea (MNU)-induced retinal detachment in C57BL6 mice	Decrease in photoreceptor cell apoptosis miR-21 upregulation [232].
RPE-Exos	N-methyl-N-nitrosourea (MNU)-induced retinal detachment in C57BL6 mice	Enhanced photoreceptor cell survival Diminished TNFα, IL-β and IL-6 mRNA expression Diminished Bax and caspase-3 mRNA expression Enhanced Bcl-2 mRNA expression [231].
	Inflammatory cytokine-induced extracellular vesicle release in ARPE-19 cell line	T cell proliferation inhibition by TNFα and IL-6 production [227].
NPC-Exos	Royal College of Surgeons rats	Decrease in photoreceptor cell apoptosis Inhibition of retinal microglia activation Diminished TNFα, IL-β and COX-2 expression [234].

## Data Availability

Not applicable.

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
