# Peer review of "Retinitis Pigmentosa: Novel Therapeutic Targets and Drug Development"

_pharmaceutics, 2023, doi:10.3390/pharmaceutics15020685_

Round 1

Reviewer 1 Report

Well organized and well written.

Author Response

Thank you for your feedback. We are thrilled to hear that you found the information to be well organized and well written. We strive to make sure our work is clear and easy to understand, so we are grateful for your positive comments. 

Reviewer 2 Report

The manuscript is well written and reports most of therapeutics that reached the preclinical and clinical stage. However, before publication, the manuscript  needs some amendments.

Paragraphs 5.2 to 5.4 must come first. Thereby, authors must report preclinical studies, and clinical data should be reported at the end of the manuscript.

Furthermore, as regards as preclinical authors must report the work authored by Enrica Strettoi and collaborators e.g. PMID: 30883241/PMID: 31199887/PMID: 20937879

Authors should try to convert some table in images i.e. table 1 and 3.

Instead figure 1 and 2 should be deleted.

Author Response

Dear Reviewer,

Thank you for taking the time to review our manuscript. We appreciate your feedback and suggestions. We have made the necessary amendments as per your request.

We appreciate your insights into the organization of our manuscript. We understand your request to move paragraphs 5.2 to 5.4 to the top of section #5, however, we would kindly request the editor's assistance in rearranging these paragraphs so that the tracking changes within these paragraphs and references are not hidden. Additionally, we believe that presenting the clinical data first aligns with the focus of our paper, which primarily highlights the preclinical data and advancements in the field. The inclusion of clinical data in sections #4 serves as a continuation and extension of our discussion on the clinical aspects of RP and conventional therapy (sections #2 and #3). Thank you for considering our perspective.

We have also made sure to include the work authored by Enrica Strettoi and collaborators, as you suggested, and have cited the relevant PMID's (30883241, 31199887, and 20937879) in the appropriate sections.

Regarding the figures 1 and 2, we understand your suggestion. While we believe that these figures are contributing to our paper, we are open to considering the editor's thoughts on their inclusion or exclusion. If the editor feels that their removal would improve the manuscript, we are willing to comply. Thank you for bringing this to our attention.

With regards to your request to convert tables 1 and 3 into images, we have considered your suggestion. However, after careful consideration, we believe that converting these tables into images may not be feasible, as they present factual information in a concise and organized manner. Instead, we have attempted to incorporate some of the information from these tables into our existing figures, which help to illustrate the mechanism of action of the therapies described.

We hope that this amendment addresses your concerns and provides clarity for our readers.

Once again, thank you for your valuable feedback. We believe that these changes have improved the overall quality of the manuscript and look forward to your positive consideration for publication.

Reviewer 3 Report

Wu K. Y, et al have reviewed and discussed the recent progress of advanced drug development for retinitis pigmentosa (RP), focused on research on molecular targets and personalized medicine at the preclinical stage. The authors discussed the underlying molecular mechanisms (pathways) which are altered and involved in RP concerning the endoplasmic reticulum (ER) stress, apoptosis, redox balance, and genomic stability.

In this study, the authors managed to demonstrate and discussed the classification, epidemiology, clinical manifestations, and prognosis of RP. The novelty of this manuscript is the discussion of the therapeutic approaches under development such as gene and cell therapy as well as the recent literature identifying novel potential drug targets for retinitis pigmentosa (RP).

Although the article provides a balanced view, the authors must discuss several points with proper references. the authors have interpreted and presented the relevant data on RP, correctly. One of the major schematics looks unattended, especially, figure 2.

Nonetheless, the article seemed to possess no major concern. Overall, the clarity of the text is good. The manuscript has a few typographical and grammatical errors. Charts and tables are much appreciated. In general, the manuscript can accomplish the caliber of quality for consideration for publication in Pharmaceutics. The authors are advised to consider the comments below:

Comments

1.      Abstract / Page 1 / line 21 / Check spelling “PR” – should be “RP”

2.      Page 3 / line 111-112 / Please provide references for this statement “Only a small 111 portion of RP patients with RPE65 gene mutation are eligible to receive the target gene 112 Therapy”

3.      Page 3 / line 131 / check spelling “Rayapudi and al 2013”

4.      Page 4 / line 140 / “vitamin A treatment remains controversial” – although a small genetic subgroup of patients with PRPH2-associated RP showed a high vitamin A treatment effect (https://iovs.arvojournals.org/article.aspx?articleid=2637214)

5.      Page 4 / 3.3 Protection from sunlight / Needed more discussion with appropriate references – (https://www.ncbi.nlm.nih.gov/pmc/articles/PMC7441298/) where they showed that an increase in housing light intensity for rd10 mice accelerates retinal degeneration, activating cell death, oxidative stress pathways.

6.      Page 4/ 3.4 Visual aids / A recent study have shown that their research device effectively assisted RP patients with night blindness (https://link.springer.com/article/10.1007/s10384-014-0354-0)

7.      Page 4 / 3.5 Surgical intervention / Safety profile of ARGUS II prosthesis is not discussed with recent finds that Argus II had a safety profile out to 4 years post-implantation and it was markedly better than observed in the pre-approval phase (https://journalretinavitreous.biomedcentral.com/articles/10.1186/s40942-021-00324-6)

8.      Page 6 / 5.1.1 Viral vectors / One of the important missing contests of AAV gene therapy is large gene delivery and/or novel capsid for AAV. Please discuss with appropriate references. (https://www.nature.com/articles/s41434-020-0174-4) & (https://www.embopress.org/doi/full/10.15252/emmm.202013392)

9.      Page 6 / 5.1.2 CRISPR-cas9 gene editing / Please discuss the RNA editing new tool – Cas13-based system (https://www.ncbi.nlm.nih.gov/pmc/articles/PMC9245891/)

10.   Page 7 / 5.1.3 RNA replacement / Please have a look at this and discuss accordingly - Mirtron-mediated RNA knockdown/replacement therapy for the treatment of dominant retinitis pigmentosa (https://www.nature.com/articles/s41467-021-25204-3)

11.   Table 1 / NRL / some of the information needs to be updated with a second reference. (https://www.ncbi.nlm.nih.gov/pmc/articles/PMC7408925/#B77-jcm-09-02224)

12.   Page 9 / The signaling pathways involved in RP is poorly explained. Please explain the transcriptional profiling and neuroprotective pathways (https://www.ncbi.nlm.nih.gov/pmc/articles/PMC8231189/).

13.   Figure 2 – consistency (the word figure 1, 2, or 3 should be written consistently either in bold or not)

14.   The schematic image (figure 2) could be a lot better drawn. Please redraw the image and take some ideas from here (https://openwetware.org/wiki/BIO254:Phototransduction).

15.   Page 15 / 6.1.2 Bone Marrow Stem Cells therapies / recent study showed that Transplanted RPCs significantly improved vision and retinal thickness as well as function in rd12 mice (https://stemcellres.biomedcentral.com/articles/10.1186/s13287-022-02828-w) – please include this study into your review.

16.   One of the important missing points is the Mechanisms of Photoreceptor Death in Retinitis Pigmentosa – via regulated necrosis (https://www.ncbi.nlm.nih.gov/pmc/articles/PMC7598671/)

Author Response

Dear Reviewer,

We are pleased that you found the manuscript well-written and informative. We have taken note of your feedback and have made the necessary amendments to the manuscript, including a proper discussion of the relevant data on retinitis pigmentosa (RP) with proper references. We have also corrected the typographical and grammatical errors and appreciate your comment on the clarity of the text.

Regarding the schematic, we have attended to the issue with figure 2 and ensured that it presents the information accurately.

All of your requests have been addressed in the revised manuscript, as shown below:

Request: Abstract / Page 1 / line 21 / Check spelling “PR” – should be “RP”

Response: Thank you for pointing this out. We have corrected the spelling from "PR" to "RP" and have also made a thorough revision of the manuscript for further mistakes.

Request: Page 3 / line 111-112 / Please provide references for this statement “Only a small 111 portion of RP patients with RPE65 gene mutation are eligible to receive the target gene 112 Therapy”

Response: Thank you for the suggestion. We have now added references to support this statement.

Request: Page 3 / line 131 / check spelling “Rayapudi and al 2013”

Response: Thank you for bringing this to our attention. We have corrected the spelling to "Rayapudi et al. (2013)".

Request: Page 4 / line 140 / “vitamin A treatment remains controversial” – although a small genetic subgroup of patients with PRPH2-associated RP showed a high vitamin A treatment effect (https://iovs.arvojournals.org/article.aspx?articleid=2637214)

Response: Thank you for the reference. We have now included this study in our discussion of the vitamin A treatment and acknowledged its potential effects on a small genetic subgroup of patients with PRPH2-associated RP.

Request: Page 4 / 3.3 Protection from sunlight / Needed more discussion with appropriate references – (https://www.ncbi.nlm.nih.gov/pmc/articles/PMC7441298/) where they showed that an increase in housing light intensity for rd10 mice accelerates retinal degeneration, activating cell death, oxidative stress pathways.

Response: Thank you for the reference. We have now added a more comprehensive discussion of protection from sunlight and included the reference in our discussion.

Request: Page 4/ 3.4 Visual aids / A recent study have shown that their research device effectively assisted RP patients with night blindness (https://link.springer.com/article/10.1007/s10384-014-0354-0)

Response: Thank you for the reference. We have now included this study in our discussion of visual aids and acknowledged its effectiveness in assisting RP patients with night blindness.

Request: Page 4 / 3.5 Surgical intervention / Safety profile of ARGUS II prosthesis is not discussed with recent finds that Argus II had a safety profile out to 4 years post-implantation and it was markedly better than observed in the pre-approval phase (https://journalretinavitreous.biomedcentral.com/articles/10.1186/s40942-021-00324-6)

Response: Thank you for the reference. We have now added a discussion of the safety profile of the ARGUS II prosthesis, including the recent findings that it had a safety profile that was significantly better than observed in the pre-approval phase.

Request: Page 6 / 5.1.1 Viral vectors / One of the important missing contents of AAV gene therapy is large gene delivery and/or novel capsid for AAV. Please discuss with appropriate references. (https://www.nature.com/articles/s41434-020-0174-4) & (https://www.embopress.org/doi/full/10.15252/emmm.202013392)

Response: Thank you for pointing out the missing information about large gene delivery and/or novel capsid for AAV in the section on viral vectors. We have reviewed the suggested references, as well as additional references, and have incorporated a discussion on this important topic in our revised manuscript.

Request: Page 6 / 5.1.2 CRISPR-cas9 gene editing / Please discuss the RNA editing new tool – Cas13-based system (https://www.ncbi.nlm.nih.gov/pmc/articles/PMC9245891/)

Response: Thank you for suggesting an additional discussion on the novel Cas13 RNA editing new tool. We have provided a brief explanation on its mechanism of action, as well as the potential limitations related to its usage.

Request: Page 7 / 5.1.3 RNA replacement / Please have a look at this and discuss accordingly - Mirtron-mediated RNA knockdown/replacement therapy for the treatment of dominant retinitis pigmentosa (https://www.nature.com/articles/s41467-021-25204-3)

Response: The information on the Mirtron-mediated RNA knockdown/replacement therapy has been considered and the relevant discussion has been added to the paper accordingly (page 7 line 291; page 13 line 470).

Request: Table 1 / NRL / some of the information needs to be updated with a second reference. (https://www.ncbi.nlm.nih.gov/pmc/articles/PMC7408925/#B77-jcm-09-02224)

Response: Thank you for your feedback. We have included the suggested reference in the text of the manuscript (page 13 line 464) since Table 1 summarizes original research papers. We have added one novel reference in Table 1 for NRL, demonstrating the importance of this gene in retinitis pigmentosa pathogenesis.

Request: Page 9 / The signaling pathways involved in RP is poorly explained. Please explain the transcriptional profiling and neuroprotective pathways (https://www.ncbi.nlm.nih.gov/pmc/articles/PMC8231189/).

Response: The signaling pathways involved in Retinitis Pigmentosa have been further explained and the transcriptional profiling and neuroprotective pathways have been added with reference to the study provided by the reviewer (page 19 line 763).

Request: Figure 2 – consistency (the word figure 1, 2, or 3 should be written consistently either in bold or not)

Response: The consistency of the labeling of figures in the paper has been checked and corrected.

Request: The schematic image (figure 2) could be a lot better drawn. Please redraw the image and take some ideas from here (https://openwetware.org/wiki/BIO254:Phototransduction).

Response: The schematic image in Figure 2 has been redrawn to better represent the information and ideas from the reference provided by the reviewer have been incorporated. We have adjusted the legend as well.

Request: Page 15 / 6.1.2 Bone Marrow Stem Cells therapies / recent study showed that Transplanted RPCs significantly improved vision and retinal thickness as well as function in rd12 mice (https://stemcellres.biomedcentral.com/articles/10.1186/s13287-022-02828-w) – please include this study into your review.

Response: The recent study on the effectiveness of bone marrow stem cell therapies has been added to the review in the section on 6.1.2 Bone Marrow Stem Cells therapies.

Request: One of the important missing points is the Mechanisms of Photoreceptor Death in Retinitis Pigmentosa – via regulated necrosis (https://www.ncbi.nlm.nih.gov/pmc/articles/PMC7598671/)

Response: An overview of the mechanism of photoreceptor death in Retinitis Pigmentosa via regulated necrosis has been added to the paper and the relevant information from the reference provided by the reviewer has been included.

We hope that these revisions address all your concerns. If there is anything else you would like us to address, please do not hesitate to let us know.

Thank you again for your time and consideration.